# DiffSDS: A geometric sequence diffusion model for protein backbone inpainting

## Abstract

Can a pure transformer learn protein structure under geometric constraints? Recent research has simplified protein structures as sequences of folding angles, making transformers suitable for unconstrained protein backbone generation. Unfortunately, such simplification is unsuitable for the constrained protein inpainting problem: we reveal theoretically that applying geometric constraints to the angle space would result in gradient vanishing or exploding, called **GradCurse**. As a remedy, we suggest adding a hidden **a**tomic **d**irection **s**pace (**ADS**) layer upon the transformer encoder, converting invariant backbone angles into equivariant direction vectors. Geometric constraints could be efficiently imposed on the direction space while avoiding GradCurse. Meanwhile, a Direct2Seq decoder with mathematical guarantees is also introduced to reconstruct the folding angles. We apply the **dual-space** model as the denoising neural network during the conditional diffusion process, resulting in a constrained generative model–**DiffSDS**. Extensive experiments show that the proposed DiffSDS outperforms the sequence diffusion baseline, and even achieves competitive results with coordinate diffusion models, filling the gap between sequence and coordinate diffusion models.

## 1 Introduction

We aim to **improve** and **simplify** the modeling of constrained protein backbone inpainting, i.e., recovering masked protein backbones, which has wide applications in *de-novo* protein design (Wang et al., 2022a; Lee & Kim, 2022; Ferruz et al., 2022). Existing protein structural generative models explicitly consider the equivariance caused by rotation and translation (Wang et al., 2022a; Trippe et al., 2022; Luo et al.; Anand & Achim, 2022). These considerations enable them to correctly consider atom interactions in the 3D space while requiring special model designs that increase the modeling complexity caused by 3D operations. Restricted by this, traditional powerful models, such as visual CNNs or sequence transformers, are prevented from being directly applied to structure modeling. Given the success of sequence transformers that seem to unify everything in NLP and CV, we wonder whether we can simplify protein backbone inpainting by treating it as a sequence modeling task. In this regard, the recent FoldingDiff (Wu et al., 2022a) suggests converting protein structures into sequences of folding angles, allowing transformers for unconditional protein backbone generation.

Unfortunately, the pure sequence model is unsuitable for constrained structure design tasks. In protein backbone inpainting, the designed structure should fit multiple geometric constraints, including linking the masked structure's endpoints and not overlapping with unmasked structures to meet the repulsion (Spassov et al., 2007; Müller-Späth et al., 2010; Drake & Pettitt, 2020). As shown in Fig.1, if the constraints are not guaranteed, specifically when the masked endpoints $s$ and $e$ are not connected, the generated structure will be meaningless. Unfortunately, we empirically find and theoretically prove that the imposing geometric constraints on the angle space generally lead to gradient explosion or vanishing called **GradCurse**. An open research question is how to efficiently and effectively impose geometric constraints on the sequence model while keeping its simplicity.

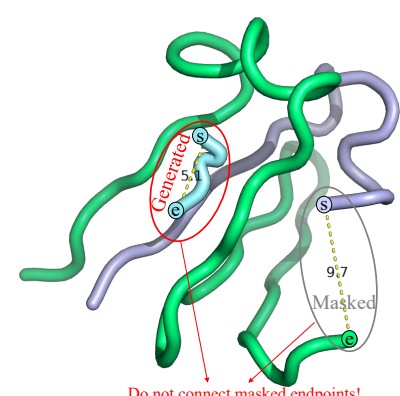

Figure 1: Violation of geometric constraints. Atoms between $s$ and $e$ are masked and should be recovered by the algorithm.

We suggest inserting a hidden **a**tomic **d**irection **s**pace (**ADS**) into the sequence model, allowing to impose structural constraints on the novel direction space efficiently. ADS is a plug-and-play cross-modal conversion technique connecting the sequence and direction space. By adding ADS upon the last transformer layer, we obtain the Seq2Direct encoder ($\text{Enc}_{\text{s2d}}$) that converts sequential features into direction vectors. We also introduce Direct2Seq decoder ($\text{Dec}_{\text{d2s}}$) according to strict mathematical transformations. A sequence model (SDS) equipped with hidden direction space could be constructed by stacking $\text{Enc}_{\text{s2d}}$ and $\text{Dec}_{\text{d2s}}$. SDS takes angular sequences as inputs and outputs, with a latent direction space that supports efficient geometric calculations and is mathematically consistent with the sequence space. In this design, multimodal constraints, e.g., sequential and 3D constraints, can be simultaneously considered on the corresponding feature space. Finally, we apply the SDS model as the denoising neural network during the conditional diffusion process, resulting in a constrained generative model–DiffSDS.

We evaluated DiffSDS on CATH4.3 and compared it with recent strong baselines, including RFDesign (Wang et al., 2022a), modified FoldingDiff (Wu et al., 2022a), SMCDiff Trippe et al. (2022), and RFDiffusion Watson et al. (2023). We also propose three metrics to evaluate the results of protein backbone inpainting, including protein likeness, connectivity, and non-overlapping. Experiments show that our methods significantly outperform baselines in all metrics. In addition, the designability of structures generated by DiffSDS is also better than baselines. As to simplicity, the proposed DiffSDS utilizes the sequence transformer to model protein structures, avoiding the equivariance consideration and GradCurse. We have also performed ablation studies to reveal the role of conditions and constraints.

## 1.1 RELATED WORK

**Problem Definition.** Protein backbone inpainting aims to recover the continuous masked substructure of the protein backbone, given the unmasked atoms as conditions. The generated structure is required to connect different protein fragments with fixed spatial positions. Formally, we write the protein backbone as $\mathcal{B} = \{\boldsymbol{p}_1^C, \boldsymbol{p}_1^A, \boldsymbol{p}_1^N, \boldsymbol{p}_2^C, \boldsymbol{p}_2^A, \boldsymbol{p}_2^N, \cdots, \boldsymbol{p}_n^C, \boldsymbol{p}_n^A, \boldsymbol{p}_n^N\}$, where $\{\boldsymbol{p}_i^C, \boldsymbol{p}_i^A, \boldsymbol{p}_i^N\}$ indicates the set of backbone atoms ($C$, $C_\alpha$ and $N$) of the $i$-th residue and $\boldsymbol{p}_i^A$ is the 3D position of the $i$-th $C_\alpha$. Denote the masked sub-structure as $\mathcal{M} = \{\boldsymbol{p}_i^C, \boldsymbol{p}_i^A, \boldsymbol{p}_i^N\}_{i=s}^e$, the unmasked structures as $\mathcal{K} = \{\boldsymbol{p}_i^C, \boldsymbol{p}_i^A, \boldsymbol{p}_i^N\}_{i=1}^{s-1} \cup \{\boldsymbol{p}_i^C, \boldsymbol{p}_i^A, \boldsymbol{p}_i^N\}_{i=e+1}^n$, where $0 < s < e < n$. We generate $\hat{\mathcal{M}}$ connecting endpoints $\boldsymbol{p}_{s-1}^N$ and $\boldsymbol{p}_{r+1}^C$ via a learnable function $f_\theta$, given $\mathcal{U}$ with a fixed conformation as input:

$$\hat{\mathcal{M}} = f_\theta(\boldsymbol{p}|\mathcal{U}, x), x \sim \mathcal{N}(0, \mathbf{I}) \tag{1}$$

Note that $\theta$ indicates learnable parameters, $x$ is a set of structural variables, such as coordinates and folding angles introduced later. The designed $\hat{\mathcal{M}}$ should be non-trivial to satisfy the following constraints:

1. Protein likeness: The designed structures are likely to constitute natural proteins.
2. Connectivity: $\hat{\mathcal{M}}$ should effectively connect $\boldsymbol{p}_{s-1}^N$ and $\boldsymbol{p}_{e+1}^C$ without breakage.
3. Non-overlapping: The designed structure $\hat{\mathcal{M}}$ should not overlap with existing structure $\mathcal{U}$.

**3D Molecule Generation.** Generating 3D molecules to explore the local minima of the energy function (Conformation Generation) (Gebauer et al., 2019; Simm et al., 2020b;a; Shi et al., 2021; Xu et al., 2021; Luo et al., 2021; Xu et al., 2020; Ganea et al., 2021; Xu et al., 2022; Hoogeboom et al., 2022; Jing et al., 2022; Zhu et al., 2022) or discover potential drug molecules binding to targeted proteins (3D Drug Design) (Imrie et al., 2020; Nesterov et al., 2020; Luo et al., 2022; Ragoza et al., 2022; Wu et al., 2022b; Huang et al., 2022a; Peng et al., 2022; Huang et al., 2022b; Wang et al., 2022b; Liu et al., 2022b) have attracted extensive attention in recent years. Compared to conformation generation that aims to predict the set of favorable conformers from the molecular graph, 3D Drug Design is more challenging in two aspects: (1) both conformation and molecule graph need to be generated, and (2) the generated molecules should satisfy multiple constraints, such as physical prior and protein-ligand binding affinity. We summarized representative works of 3D drug design in Table.5 in the appendix, where all the methods focus on small molecule design.

**Protein Design.** In addition to small molecules, biomolecules such as proteins have also attracted considerable attention from researchers (Ding et al., 2022; Ovchinnikov & Huang, 2021; Gao et al., 2020; Strokach & Kim, 2022). We divide the mainstream protein design methods into three categories: protein sequence design (Li et al., 2014; Wu et al., 2021; Pearce & Zhang, 2021; Ingraham et al., 2019;

Jing et al., 2020; Tan et al., 2022; Gao et al., 2022a; Hsu et al., 2022; Dauparas et al., 2022; Gao et al., 2022b; O'Connell et al., 2018; Wang et al., 2018; Qi & Zhang, 2020; Strokach et al., 2020; Chen et al., 2019; Zhang et al., 2020; Anand & Achim, 2022), unconditional protein structure generation (Anand & Huang, 2018; Sabban & Markovsky, 2020; Eguchi et al., 2022; Wu et al., 2022a), and conditional protein design (Lee & Kim, 2022; Wang et al., 2022a; Trippe et al., 2022; Lai et al., 2022; Fu & Sun, 2022; Tischer et al., 2020; Anand & Achim, 2022; Luo et al.). Protein sequence design aims to discover protein sequences folding into the desired structure, and unconditional protein structure generation focuses on generating new protein structures from noisy inputs. We are interested in conditional protein design and consider multiple constraints on the designed protein. For example, Wang's model (Wang et al., 2022a), SMCDiff (Trippe et al., 2022) and Tischer's model (Tischer et al., 2020) design the scaffold for the specified functional sites. ProteinSGM (Lee & Kim, 2022) mask short spans ($< 8$ residues) of different secondary structures in different structures and treats the design task as an inpainting problem. CoordVAE (Lai et al., 2022) produces novel protein structures conditioned on the backbone template. RefineGNN (Jin et al., 2021), CEM (Fu & Sun, 2022), and DiffAb (Luo et al.) aim to generate the complementarity-determining regions of the antibody. We summarized the protein design model in Table.6.

**Sequence Diffusion for Protein Structure Generation.** Diffusion models (Sohl-Dickstein et al., 2015; Ho et al., 2020; Cao et al., 2022) are a class of generative models that have achieved impressive results in image (Song et al., 2020; Lugmayr et al., 2022; Whang et al., 2022; Baranchuk et al., 2021; Wolleb et al., 2022), speech (Lee & Han, 2021; Chen et al., 2020; Kong et al., 2020; Liu et al., 2022a) and text (Li et al., 2022; Chen et al., 2022; Austin et al., 2021) synthesis. Recently, FoldingDiff (Wu et al., 2022a) shows that sequence models could be used for unconditional protein generation.

## 2 BACKGROUND AND KNOWLEDGE GAP

Considering three backbone atoms $(N, C_\alpha, C)$ for each residue, there are $9 (= 3 \times 3)$ freedom degrees are required for 3D representation. Recently, researchers have introduced human knowledge into protein backbone representation and proposed two simplified approaches: frame-based and angle-based representation, as shown in Fig.2 (a).

**Frame-based.** This approach (Jumper et al., 2021) treats residues as fundamental elements and assumes that residues of the same type have the same rigid structure, called the local frame. As shown in Fig.2 (a), we write $F_i = \{C_i, C_{\alpha_i}, N_i\}$ as the local frame of the $i$-th residue, where position $\boldsymbol{p}_{F_i}$, orientation $R_i$ and the residue type $s_i$ are required for describing $F_i$, resulting in $7(= 3 + 3 + 1)$ freedom degrees. Under this representation, geometric features can be computed efficiently, e.g., pairwise distance and relative positions. However, the model needs to consider the geometric equivariance of the input data, which introduces considerable modeling complexity.

**Angle-based.** This approach converts structures into sequences of backbone angles based on the order of the protein's primary structure; see Fig.2 (a). By assuming the backbone bond lengths are fixed, three bond angles $\alpha_i^N, \alpha_i^A, \alpha_i^C$ and three torsion angles $\beta_i^N, \beta_i^A, \beta_i^C$ are required for describing one residue, leading to 6 freedom degrees. The reduced freedom forms a more compact representation than the frame-based approach. In addition, there is no need to consider geometric equivariance since all angles are invariant to spatial rotation and translation.

**Geometric Constraints.** The angle-based representation seems attractive for simplifying structural modeling and learning more compact protein representations. However, it suffers from the drawback of inefficient computing of geometric features, making it challenging to consider geometric constraints. For example, if one wants to optimize

$$\mathcal{L}_{dist} = \min_{\{\alpha_i^N, \alpha_i^A, \alpha_i^C, \beta_i^N, \beta_i^A, \beta_i^C\}_0^i} (||\boldsymbol{p}_i^A - \boldsymbol{p}_1^A|| - r)^2 \tag{2}$$

given $\boldsymbol{p}_1^N, \boldsymbol{p}_1^A, \boldsymbol{p}_1^C$. Then, $\boldsymbol{p}_2^N \to \boldsymbol{p}_2^A \to \boldsymbol{p}_2^C \to \boldsymbol{p}_3^N \to \boldsymbol{p}_3^A \to \cdots \to \boldsymbol{p}_i^A$ needs to be recursively computed by

$$\begin{cases} \boldsymbol{p}_i^N = \text{Place}(\boldsymbol{p}_{i-1}^C, \alpha_{i-1}^N, \beta_{i-1}^N, \boldsymbol{d}_{i-1}^C, \boldsymbol{d}_{i-1}^A, r^N) \\ \boldsymbol{p}_i^A = \text{Place}(\boldsymbol{p}_i^N, \alpha_i^A, \beta_{i-1}^A, \boldsymbol{d}_i^N, \boldsymbol{d}_{i-1}^C, r^A) \\ \boldsymbol{p}_i^C = \text{Place}(\boldsymbol{p}_i^A, \alpha_i^C, \beta_i^C, \boldsymbol{d}_i^A, \boldsymbol{d}_i^N, r^C) \end{cases} \tag{3}$$

where $\boldsymbol{d}_i^N = \frac{\boldsymbol{p}_i^N - \boldsymbol{p}_{i-1}^C}{||\boldsymbol{p}_i^N - \boldsymbol{p}_{i-1}^C||}, \boldsymbol{d}_i^A = \frac{\boldsymbol{p}_i^A - \boldsymbol{p}_i^N}{||\boldsymbol{p}_i^A - \boldsymbol{p}_i^N||}, \boldsymbol{d}_i^C = \frac{\boldsymbol{p}_i^C - \boldsymbol{p}_i^A}{||\boldsymbol{p}_i^C - \boldsymbol{p}_i^A||}$. Note that backbone bond lengths, e.g., $r^N = ||\boldsymbol{p}_i^N - \boldsymbol{p}_{i-1}^C||, r^A = ||\boldsymbol{p}_i^A - \boldsymbol{p}_i^N||, r^C = ||\boldsymbol{p}_i^C - \boldsymbol{p}_i^A||$, are constants. Alg.1 (in the appendix) shows the details of place$(\boldsymbol{p}, \alpha, \beta, \boldsymbol{d}_1, \boldsymbol{d}_2)$.

**GradCurse.** The aforementioned symbols $N$, $A$, and $C$ are used to identify the backbone atoms. To simplify notation, let's rewrite $\{\boldsymbol{p}_1^A, \boldsymbol{p}_1^C, \boldsymbol{p}_2^N, \boldsymbol{p}_2^A, \boldsymbol{p}_2^C, \cdots, \boldsymbol{p}_i^N, \boldsymbol{p}_i^A\}$ as $\{\boldsymbol{x}_0, \boldsymbol{x}_1, \boldsymbol{x}_2, \boldsymbol{x}_3, \cdots, \boldsymbol{x}_{n-1}, \boldsymbol{x}_n\}$. We define $\boldsymbol{d}_i = \frac{\boldsymbol{x}_i - \boldsymbol{x}_{i-1}}{||\boldsymbol{x}_i - \boldsymbol{x}_{i-1}||}$, $r_i = ||\boldsymbol{x}_i - \boldsymbol{x}_{i-1}||$, $\boldsymbol{e} = \frac{\boldsymbol{x}_n - \boldsymbol{x}_0}{||\boldsymbol{x}_n - \boldsymbol{x}_0||}$, and $L = \sum_{i=1}^n r_i \boldsymbol{e}^T \boldsymbol{d}_i$. Consequently, Equation 2 can be reformulated as $\mathcal{L}(\sum_{i=1}^n r_i \boldsymbol{e}^T \boldsymbol{d}_i; r)$. As proved in the appendix, under mild assumptions, the projected gradient along the $\boldsymbol{e}$ direction is:

$$\frac{\partial \mathcal{L}}{\partial \boldsymbol{d}_i} \boldsymbol{e} = \frac{\partial \mathcal{L}}{\partial L}(\sum_{k=i}^n r_k \boldsymbol{e}^T \frac{\partial \boldsymbol{d}_k}{\partial \boldsymbol{d}_i} \boldsymbol{e}) \tag{4}$$

where the expected value of $\boldsymbol{e}^T \frac{\partial \boldsymbol{d}_n}{\partial \boldsymbol{d}_{n-k}} \boldsymbol{e}$ could be estimated as:

$$\mathbb{E}[\boldsymbol{e}^T \frac{\partial \boldsymbol{d}_n}{\partial \boldsymbol{d}_{n-k}} \boldsymbol{e}] \approx K_1 t_1^k + K_2 t_2^k \tag{5}$$

Here, $K_1, K_2, t_1, t_2$ are constants. The Eq.5 shows that the projected gradient is either exploding or vanishing, resulting in unstable training. We observe this phenomenon in our experiments and call this phenomenon **GradCurse** of angle-based representation methods.

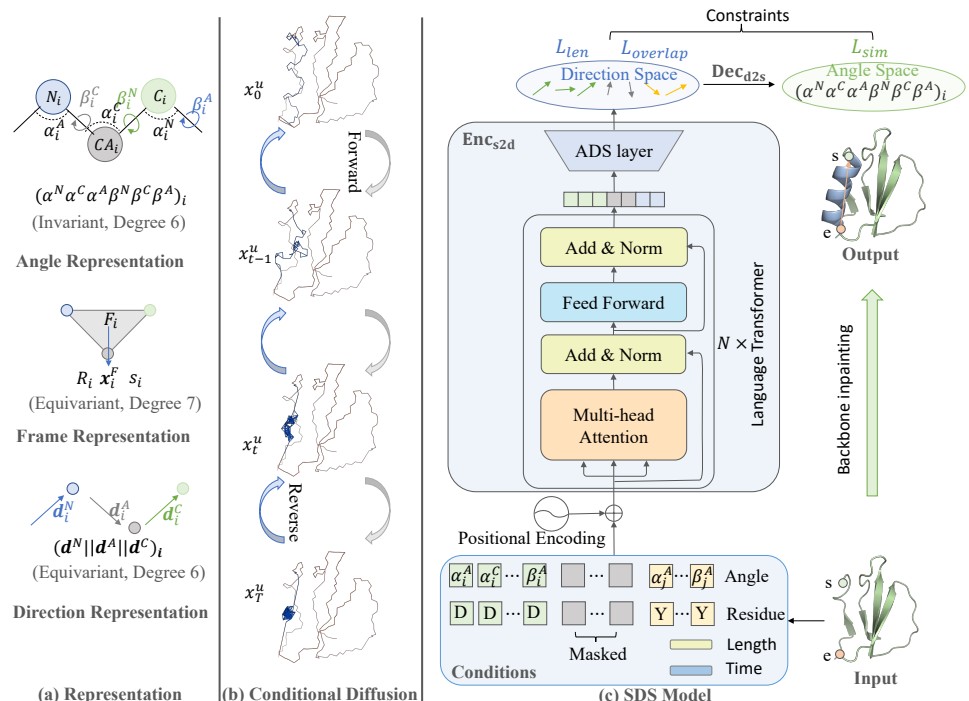

Figure 2: The overall framework. (a) We introduce a new direction representation for efficiently computing geometric features while enjoying minimal freedom. (b) Only the masked structures are involved in the diffusion process, while the unmasked ones remain fixed. (c) We add an ADS layer upon the sequence transformer to convert invariant features into equivariant directions, namely $\text{Enc}_{\text{s2d}}$. Then the $\text{Dec}_{\text{d2s}}$ reverses directions as invariant features based on Eq.17. The geometric constraints ($\mathcal{L}_{len}$ and $\mathcal{L}_{overlap}$) could be efficiently imposed on the direction space. The overall $\text{Enc}_{\text{s2d}} + \text{Dec}_{\text{d2s}}$ is a simple sequence model. Remind that the term "mask" may be abused here as we corrupt the angles with wrapped Gaussian noise instead of replacing them with a special mask token.

## 3 METHOD

### 3.1 OVERALL FRAMEWORK & NOVEL REPRESENTATION

We introduce a novel direction-based representation for protein backbone, which enjoys the simplicity of angle-based representations while effectively mitigating the issue of GradCurse. Based on the direction-based representation, we propose a sequence diffusion model for constrained protein backbone inpainting, called DiffSDS. The model takes unmasked atoms and controllable conditions as input to recover the masked region, as shown in Fig.2.

**Direction-based Representation.** Is there an alternative representation beyond frame- and angle-based ones to support efficient computation of geometric features while enjoying low degrees of freedom? As shown in Fig.2(a), we introduce direction vectors, i.e., $d_i^A, d_i^N, d_i^C$, for discribing protein structures. In the direction-based representation, the position of each atom is determined by its relative direction $(d_i^A, d_i^N, d_i^C)$ and distance $(r^N, r^A, r^C)$ from its parent node. In Fig.2(a), taking $C_\alpha$ as an example,

$$p_i^A = p_i^N + r^A d_i^A \qquad (6)$$

Recall that $p_i^A$ and $p_i^N$ are spatial coordinates of $C_{\alpha_i}$ and $N_i$. The direction vector $d_i^A$ points from $p_i^N$ to $p_i^A$, and $r^A$ is the length of the $C_\alpha - C$ bond.

**Advantages.** The proposed representation has several advantages. Firstly, the computing cost of relative positions will be reduced. For example, when computing $p_i^A - p_0^A$, only parallel linear additions and multiplications are required, without recursive computation as in Eq.3:

$$p_i^A - p_0^A = r^A \sum_{k=2}^{i} d_i^A + r^C \sum_{k=1}^{i-1} d_i^C + r^N \sum_{k=2}^{i} d_i^N \qquad (7)$$

Secondly, there is no gradient vanishing or exploding issue in Eq.7. Thirdly, this representation enjoys the lowest 6 freedom degrees since $||d_i^A|| = ||d_i^C|| = ||d_i^N|| = 1$. Finally, direction representation could be equivalently transformed to the angle-based one, as shown in Eq.17 in the appendix.

### 3.2 DUAL-SPACE DIFFUSION MODEL

We propose a sequence diffusion model $f_\theta$ equipped with direction space for recovering masked backbone $\mathcal{M}$ conditional on the unmasked part $\mathcal{U}$:

$$\hat{\mathcal{M}} = f_\theta(p|\mathcal{U}, x), x \sim \mathcal{N}(0, \mathbf{I}) \qquad (8)$$

$f_\theta$ consists multiple bert transformer layers, to which we add an ADS layer, i.e., a linear projection head $\text{ADS}(h) : h \in \mathbb{R}^{n,d_h} \to d \in \mathbb{R}^{n,3}$, to convert hidden feature vectors $h$ as direction vectors $d$.

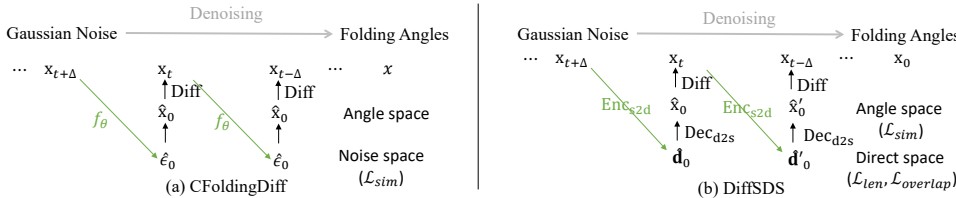

Figure 3: Reverse diffusion. (a) FoldingDiff uses the neural model $f_\theta$ to predict one-step noise, where angular similarity loss is imposed on the noise space. We extend FoldingDiff to conditional scenes by fixing the unmasked structure during diffusion to obtain CFoldingDiff. (b) DiffSDS uses $\text{Enc}_{s2d}$ to predict the direction representation of the original protein. The angular similarity loss and geometric constraints are imposed in the angle and direction space, respectively.

**Conditions.** The model takes multiple prior conditions as input features for each residue, as shown in Fig.2(c). The node-level conditional features include backbone angles $F_{ang} \in \mathbb{R}^{n,6}$, and residue type embedding $F_{res} \in \mathbb{R}^{n,20}$ of unmasked residues. The global-level conditional features include the length $(F_{len} \in \mathbb{R})$ of the masked fragment and diffusion timestamp $(F_{time} \in \mathbb{R})$.

**Conditional Forward Diffusion.**  The forward diffusion process could be viewed as a mixup path from clean data $x_0 \sim p_{data}$ to noise $x_T \sim \mathcal{N}(0, I)$: $x_0 \rightarrow x_1 \rightarrow \cdots \rightarrow x_T$. Different from generating proteins from scratch, the structure $\mathcal{U}$ is given as a prior, whose angles should not be changed during the diffusion process. Therefore, we divide the latent variable $x_t$ into two parts: $x_t = x_t^m \bigoplus x_t^u$, where $x_t^m$ and $x_t^u$ are the masked and unmasked protein angles at timestamp $t$. Denote $\alpha_0 = 1, \sigma_0 = 0, q(x_0|x_0) = \mathcal{N}(x_0; \alpha_0 x, \sigma_0^2 I)$, we have

$$q(x_t^m|x_0^m) = \mathcal{N}_{\text{wrapped}}(x_t^m; \alpha_t x_0^m, \sigma_t^2 I) \propto \sum_{k=-\infty}^{\infty} \exp\left(\frac{-||x_t^m - \alpha_t x_0^m + 2\pi k||}{2\sigma_t^2}\right) \quad (9)$$

where we use the wrapped normal (Wu et al., 2022a) to force the angles space in $[0, \pi]$. The hyper-parameters $\alpha_t$ and $\sigma_t$ determine the diffusion schedule, i.e., $\sigma_t = \text{clip}(1 - \alpha_t, 0.999)$ and $\alpha_t = \cos\left(t/T \cdot \frac{\pi}{2}\right)$.

**Direction-aware Reverse Diffusion.**  The reverse process applies neural network $f_\theta$ as the translation kernel to recover clean data following the Markov chain $x_T \rightarrow x_{T-1} \rightarrow \cdots \rightarrow x_0$. As derived in the Appendix, the objective is to maximize $q(x_{t-1}|x_t, x_0) = \mathcal{N}(z_{t-1}; \hat{\mu}_{t-1}, \hat{\sigma}_{t-1}^2 I)$, and

$$\begin{cases} \hat{\sigma}_s = \frac{\sigma_{t|s}\sigma_s}{\sigma_t} \\ \hat{\mu}_s = \frac{1}{\alpha_{t|s}}x_t - \frac{\alpha_{t|s}}{\sigma_t}\epsilon_t \end{cases} \quad (10)$$

There are several alternative variables could be estimated by $f_\theta$ to obtain $q(x_{t-1}|x_t, x_0)$, such as $\hat{\mu}_{t-1}, \hat{\epsilon}_{t-1}, \hat{x}_0$. As shown in Fig.3, FoldingDiff realizes the neural network as $f_\theta : (x_t, t) \mapsto \hat{\epsilon}_t$, which is effective for unconditional protein backbone generation. Instead, we prefer $f_\theta : (x_t, t) \mapsto \hat{x}_0$ and decompose it as encoder $\text{Enc}_{\text{s2d}}$ and decoder $\text{Dec}_{\text{d2s}}$. The $\text{Enc}_{\text{s2d}} : (x_t, t) \mapsto \hat{d}_0$ predicts the direction vectors $(d_0)$ of the backbone, and the decoder $\text{Dec}_{\text{d2s}} : \hat{d}_0 \mapsto \hat{x}_0$ reverses the direction representation into angle representation based on Eq.17. With the inserted direction space, we could efficiently compute geometric features from and impose corresponding constraints on the model, as illustrated in Eq.6 and Eq.2. More importantly, this modification does not increase the modeling complexity: $f_\theta$ still appears as a sequence model, with the inputs and outputs being sequences.

### 3.3 CONSTRAINTS

**Protein Likeness.**  From the diffusion perspective, the neural model needs to recover the masked backbone angles to ensure the protein likeness. The overall objective is to maximize the variational lower bound of $\log p_\theta(x_0)$:

$$\mathcal{L}_{vlb}(x_0) = \mathbb{E}_{q(x_{1:T}|x_0)}\left[\log\frac{q(x_T|x_0)}{p_\theta(x_T)} + \sum_{t=2}^{T}\log\frac{q(x_{t-1}|x_0, x_t)}{p_\theta(x_{t-1}|x_t)} - \log p_\theta(x_0|x_1)\right] \quad (11)$$

In practice, we use the simplfied loss: $\mathcal{L}_{sim}(x_0) = \sum_{t=0}^{T}||f_\theta(x_t, t) - x_0||^2$.

**Length Loss.**  To ensure the designed $\hat{\mathcal{M}}$ has a similar length as the reference structure $\mathcal{M}$, such that the masked endpoints ($s$ and $e$) could be connected, we employ the following loss on the direction space:

$$\mathcal{L}_{len} = \sum_{S\in\{N,A,C\}}(||\hat{\boldsymbol{p}}_s^S - \hat{\boldsymbol{p}}_e^S|| - ||\boldsymbol{p}_s^S - \boldsymbol{p}_e^S||)^2 \quad (12)$$

where $\hat{\boldsymbol{p}}_s^S - \hat{\boldsymbol{p}}_e^S$ are computed by Eq.7 using the output directions of $\text{Enc}_{\text{s2d}}$.

**Overlapping Loss.**  To avoid overlapping between designed $\hat{\mathcal{M}}$ and unmasked structure $\mathcal{U}$, the overlapping loss is also imposed on the direction space:

$$\mathcal{L}_{overlap} = \sum_{i\in[s+1,e-1]} e^{-\tau||\hat{\boldsymbol{p}}_i^A - \boldsymbol{p}_j^A||} \quad (13)$$

where $j = \min_{j\in[1,s]\cup[e,n]}||\boldsymbol{p}_i^A - \boldsymbol{p}_j^A||$ is the nearst $\alpha$-carbon atom to $\hat{\boldsymbol{p}}_i^A$ in $\mathcal{U}$. We set $\tau = 0.8$.

**Overall Loss.** During training, we impose protein similarity loss ($\mathcal{L}_{sim}$), length loss ($\mathcal{L}_{len}$), and overlapping loss ($\mathcal{L}_{overlap}$) on the model, the overall loss function is:

$$\mathcal{L} = \lambda_1\mathcal{L}_{sim} + \lambda_2\mathcal{L}_{len} + \lambda_3\mathcal{L}_{overlap} \tag{14}$$

where we choose $\lambda_1 = 1, \lambda_2 = 0.001, \lambda_3 = 10$.

## 4 EXPERIMENTS

In this section, we conduct experiments to answer the following questions:

- **Q1: Comparision.** Could DiffSDS outperform the angle-based CFoldingDiff and even demonstrate competitive results comparable to frame-based models?
- **Q2: Ablation.** Does the conditional features and constraints improves performance?

### 4.1 OVERALL SETTING

**Data Split.** We train models on CATH4.3, where proteins are partitioned by the CATH topology classification. To avoid potential information leakage, we further refine the test set by excluding proteins that are similar to the training data from the test set, i.e., TM-score greater than 0.5. Finally, there are 24,199 proteins for training, 3,094 proteins for validation, and 378 proteins for testing.

**Baselines.** We compare DiffSDS with recent sequence modeling baselines (CFoldingDiff) and coordinates diffusion baselines (SMCDiff (Trippe et al., 2022), RFDesign (Wang et al., 2022a), and RFDiffusion (Watson et al., 2023)). CFoldingDiff is a derivative of FoldingDiff (Wu et al., 2022a) where the angles and residue types of the unmasked residues are fixed during diffusion. RFDesign and RFDiffusion are state-of-the-art structural models trained across the whole PDB dataset and accepted by Science and Nature, respectively. While there are other interesting protein structure generative models, such as IG-VAE (Eguchi et al., 2022), Genie (Lin & AlQuraishi, 2023), and FrameDiff (Yim et al., 2023), it is important to note that they focus on unconditional generation instead of conditional inpainting.

**Setting.** We evaluate all methods on the same test dataset, where the contiguous backbones of length $m \sim U(5, L/3)$ are randomly masked, given the protein length $L$. For the same protein, the masked area remains the same when evaluating different methods. We retrain CFoldingDiff to make it suitable for the inpainting task, while the pre-trained SMCDiff, RFDesign and RFDiffusion can be used directly. For DiffSDS, we utilize 16 transformer layers with 384 hidden dimensions and 12 attention heads per layer. CFoldingDiff and DiffSDS are trained for up to 10,000 epochs with an early stopping patience of 1,000. The learning rate is set to 0.0001, the batch size to 128, and the maximum diffusion timestamp $T$ to 1,000.

### 4.2 PROTEIN LIKENESS

**Objective & Setting.** Are the designed structures likely to constitute native proteins? We take Rosetta energies (rama and omega) as metrics to measure the protein likeness of the generated backbones. The "rama" indicates Ramachandran torsion energy derived from statistics on the PDB, and "omega" indicates omega angle energy. In the appendix, we show the angle distributions and Ramachandran plots of the different methods in Fig.6. We group results by the masked length to reveal the performance at different mask lengths. We also adopt the energy of the test set structures as a baseline to show how closely we approximate the reference structures.

| Representation | energy | rama ($\downarrow$) | | | omega ($\downarrow$) | | |
|---|---|---|---|---|---|---|---|
| | mask length | <15 | 15-30 | >30 | <15 | 15-30 | >30 |
| | Test | 0.67 | 0.71 | 0.62 | 0.75 | 0.68 | 0.66 |
| Frame-based | RFDesign | 2.12 | 2.49 | 3.38 | 16.62 | 12.56 | 13.14 |
| | RFDiffusion | **1.19** | **1.11** | **0.87** | **1.36** | **1.2** | **1.21** |
| Angle-based | CFoldingDiff | 1.65 | 1.86 | 2.11 | 6.26 | 4.97 | 4.17 |
| | DiffSDS | **1.51** | **1.76** | **1.95** | **4.30** | **3.17** | **2.77** |

Table 1: Rosetta energies of generated backbones. We highlight the **best** results within each group.

**Results & Analysis.** We present results in Table.1. DiffSDS outperforms angle-based CFoldingDiff and provides competitive results compared to frame-based methods. The improvements to CFoldingDiff are consistent in terms of different masked lengths. However, we also observe that there still exists a considerable performance gap between the angle-based methods and the best frame-based RFDiffusion, where DiffSDS is the first attempt to fill this gap. In Figure.6, we compare the angular distributions of different methods, where DiffSDS's results are close to the test set distribution.

## 4.3 CONNECTIVITY

**Objective & Setting.** Do the designed structures connect the endpoints without breakage? As shown in Figure.1, the connectivity is an important indicator for the inpainting task, as disconnected proteins must be structurally abnormal. However, this metric has lacked attention in previous work, and we begin by defining the connectivity error as:

$$\text{Error} = ||\boldsymbol{p}_s^A - \hat{\boldsymbol{p}}_s^A|| + ||\boldsymbol{p}_e^A - \hat{\boldsymbol{p}}_e^A|| \tag{15}$$

where $s$ and $e$ are indexes of the start and end points of the masked structure, $\hat{\boldsymbol{p}}^A$ and $\boldsymbol{p}^A$ indicate the predicted and ground truth positions of the $C_\alpha$-carbon. Ideally, Error $= 0$ means that endpoints are connected as expected.

**Results & Analysis.** From Table.2, we conclude that DiffSDS could achieve the lowest connectivity error compared to baselines. We identify that good connectiveness is a significant difference made by DiffSDS, as explicitly imposing geometric constraints on the direction space could avoid structural breaks. Despite the protein-like appearance of the generated fragment, previous frame-based methods fail to ensure connectiveness, resulting in an overall flawed inpainted protein structure. We further show the trend of connectivity error with increasing mask length in Figure.7, from which we find that SMCDiff and RFDesign perform poorly at all mask lengths, FoldingDiff's connectivity error increases with mask length, while DiffSDS performs steadily and consistently better than all baselines.

| Representation | | Connectivity error ($\downarrow$) | | |
|---|---|---|---|---|
| | mask length | <10 | 10-15 | >15 |
| Frame-based | SMCDiff | **113.62** | **107.32** | **154.52** |
| | RFDesign | 218.36 | 146.24 | 159.17 |
| | RFDiffusion | 213.21 | 141.99 | 158.67 |
| Angle-based | CFoldingDiff | 14.77 | 42.65 | 59.08 |
| | DiffSDS | **6.93** | **9.93** | **10.61** |

Table 2: Connectivity error of different methods.

## 4.4 NON-OVERLAPPING

**Objective & Setting.** Will the designed structures be overlapped with existing backbones? We evaluate the spatial interaction between the generated structure ($\hat{\mathcal{M}}$) and the unmasked structure ($\mathcal{U}$), and define the interaction score as

$$\text{Score}_d = \sum_{i \in \hat{\mathcal{M}}, j \in \mathcal{U}} \mathbb{1}(||\hat{\boldsymbol{x}}_i - \hat{\boldsymbol{x}}_j|| < d) \tag{16}$$

where $\mathbb{1}(\cdot)$ is an indicator function. $\text{Score}_d$ records the number of pairwise interactions between masked and non-masked amino acids, with distances threshold $d$ $\mathring{A}$.

| Representation | | $d = 1\mathring{A}$ | | | $d = 3\mathring{A}$ | | |
|---|---|---|---|---|---|---|---|
| | mask length | <15 | 15-30 | >30 | <15 | 15-30 | >30 |
| | True | 266 | 345 | 139 | 268 | 406 | 139 |
| Frame-based | RFDesign | 280 | 357 | 153 | 861 | 1362 | 593 |
| | RFDiffusion | **266** | **345** | **139** | 275 | **347** | **139** |
| Angle-based | CFoldingDiff | 276 | **352** | 145 | 552 | 640 | 222 |
| | DiffSDS | **270** | 356 | **141** | 472 | **632** | **178** |

Table 3: The number of spatial interactions.

**Results & Analysis.** As shown in Table.3, the spatial interactions of DiffSDS's generated structures are closer to the test set than the CFoldingDiff. This verifies that incorporating a non-overlap loss in the direction space can be beneficial for angle-based methods in generating reasonable structures.

### 4.5 ABLATION STUDY

**Objective&Setting.** We conduct ablation experiments to investigate the impact of conditions and constraints on protein backbone inpainting. Specifically, we show how these factors affect the validation losses, including angle loss, length loss, and overlapping loss.

**Results & Analysis.** Limited to the page, we show the ablation details in the Appendix. As shown in Fig.8 and Fig.9, we find that: (1) the length condition and the sequence condition contribute to the reduction of $\mathcal{L}_{len}$ and $\mathcal{L}_{sim}$, respectively. This phenomenon suggests that the model can learn to generate structures with a predetermined length and that residue types can facilitate learning backbone angles. (2) Explicitly imposing geometric constraints on the model is necessary. If the constraints are removed, the $\mathcal{L}_{len}$ loss is difficult to reduce, and the $\mathcal{L}_{overlap}$ even increases, indicating the model could not generate geometrically reasonable structures. Fortunately, all these drawbacks could be eliminated by imposing geometric loss on the introduced direction space.

### 4.6 DESIGNABILITY

**Objective & Setting.** How likely are the generated proteins to be synthesized in the laboratory? We further measure the designability of the generated protein by the self-consistency TM score (scTM), which is first introduced by FoldingDiff. The generated backbones are fed into ESM-IF to obtain candidate protein sequences, which are subsequently folded into 3D structures by OmegaFold. scTM is the TM score between the newly folded structure and the original generated structure. If scTM$> 0.5$, the corresponding generated backbone is considered designable.

**Results & Analysis.** As shown in Table4, the designability order is: RFDiffusion $>$ DiffSDS$>$CFoldingDiff$>$RFDesign$>$SMCDiff. On the full test set, DiffSDS generates 231 designable backbones, outperforming CFoldingDiff's 217 and RFDesign's 178. Compared with CFoldingDiff, DiffSDS achieves consistent improvements in all settings. The pure transformer-based DiffSDS, without considering the equivariance, even outperforms several frame-based methods, such as SMCDiff and RFDesign. Despite the gap from RFDiffusion is still large, DiffSDS presents a new modeling paradigm and promising results for protein structure generation. In Fig.4, we show a protein inpainting example, comparing different methods. Combine with Table.2, we argue that structural breaks due to the lack of structural constraints would corrupt the input features of ESM-IF for SMCDiff and RFDesign, producing poor sequence design and ultimately poor scTM scores.

| Representation | | scTM>0.5 (↑) | | | scTM (↑) | |
|---|---|---|---|---|---|---|
| | | All | len≤70 | len>70 | Mean | Median |
| Frame-based | SMCDiff | 30/378 | 12/148 | 18/230 | 0.36 | 0.34 |
| | RFDesign | 178/378 | 65/148 | 113/230 | 0.51 | 0.48 |
| | RFDiffusion | **255/378** | **103/148** | **152/230** | **0.60** | **0.59** |
| Angle-based | CFoldingDiff | 217/378 | 81/148 | 130/230 | 0.54 | 0.53 |
| | DiffSDS | **231/378** | **88/148** | **143/230** | **0.56** | **0.55** |

Table 4: Designability of different methods.

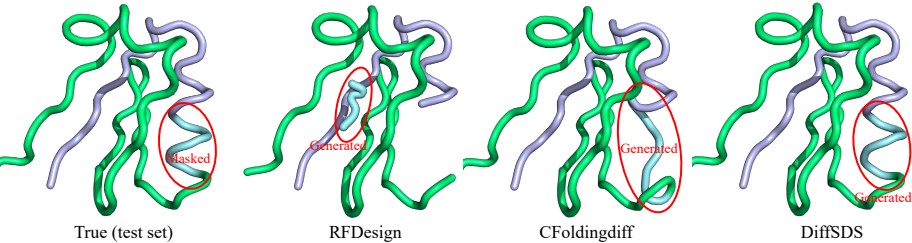

True (test set)     RFDesign     CFoldingdiff     DiffSDS

Figure 4: Inpating examples. DiffSDS generates a more similar structure to the reference protein.

## 5 CONCLUSION

This paper enhances the pure-transformer as a strong structure learner by introducing a hidden direction-based space. The proposed method not only maintains the simplicity of sequence modeling but also enables efficient and effective computation of geometric features, avoiding the issue of GradCurse. By employing the model within the framework of conditional diffusion, the proposed DiffSDS bridges the gap between angle-based and frame-based diffusion models, surpassing the angle-based CFoldingDiff method, and introducing a new paradigm for protein structure learning.

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
