# A APPENDIX

## A.1 RELATED WORKS

**3D Molecule Generation.** Generating 3D molecules to explore the local minima of the energy function (Conformation Generation) Gebauer et al. (2019); Simm et al. (2020b;a); Shi et al. (2021); Xu et al. (2021); Luo et al. (2021); Xu et al. (2020); Ganea et al. (2021); Xu et al. (2022); Hoogeboom et al. (2022); Jing et al. (2022); Zhu et al. (2022) or discover potential drug molecules binding to targeted proteins (3D Drug Design) Imrie et al. (2020); Nesterov et al. (2020); Luo et al. (2022); Ragoza et al. (2022); Wu et al. (2022b); Huang et al. (2022a); Peng et al. (2022); Huang et al. (2022b); Wang et al. (2022b); Liu et al. (2022b) have attracted extensive attention in recent years. Compared to conformation generation that aims to predict the set of favourable conformers from the molecular graph, 3D Drug Design is more challenging in two aspects: (1) both conformation and molecule graph need to be generated, and (2) the generated molecules should satisfy multiple constraints, such as physical prior and protein-ligand binding affinity. We summarized representative works of 3D drug design in Table.5 in the appendix, where all the methods focus on small molecule design.

Table 5: 3D molecule generation models.

| Method | Input | Github |
|---|---|---|
| *Molecule Conformation Generation* | | |
| G-SchNet Gebauer et al. (2019) | – | PyTorch |
| CVGAE Mansimov et al. (2019) | 2D-graph | TF |
| GraphDG Simm et al. (2020b) | 2D-graph | PyTorch |
| MolGym Simm et al. (2020a) | – | PyTorch |
| ConfGF Shi et al. (2021) | 2D-graph | PyTorch |
| ConfVAE Xu et al. (2021) | 2D-graph | PyTorch |
| DGSM Luo et al. (2021) | 2D-graph | – |
| CGCF Xu et al. (2020) | 2D-graph | PyTorch |
| GeoMol Ganea et al. (2021) | 2D-graph | PyTorch |
| G-SphereNet Luo & Ji (2021) | – | PyTorch |
| GeoDiff Xu et al. (2022) | 2D-graph | PyTorch |
| EDM Hoogeboom et al. (2022) | 2D-graph | PyTorch |
| TorsionDiff Jing et al. (2022) | 2D-graph | PyTorch |
| DMCG Zhu et al. (2022) | 2D-graph | PyTorch |
| *De novo* Molecule Design | | |
| DeLinker Imrie et al. (2020) | Protein Pocket 3D-fragments | TF |
| 3DMolNet Nesterov et al. (2020) | 3D-geometry | – |
| cG-SchNet Gebauer et al. (2022) | 3D-geometry | PyTorch |
| Luo's model Luo et al. (2022) | Protein Pocket | PyTorch |
| LiGAN Ragoza et al. (2022) | Protein Pocket | PyTorch |
| Bridge Wu et al. (2022b) | Physical prior | – |
| MDM Huang et al. (2022a) | 2D-graph Properties | – |
| Pocket2Mol Peng et al. (2022) | Protein Pocket | PyTorch |
| 3DLinkcer Huang et al. (2022b) | 3D-fragments | PyTorch |
| CGVAE Wang et al. (2022b) | Coarse Topology | PyTorch |
| GraphBP Liu et al. (2022b) | Protein Pocket | PyTorch |

**Protein Design.** In addition to small molecules, biomolecules such as proteins have also attracted considerable attention by researchers (Ding et al., 2022; Ovchinnikov & Huang, 2021; Gao et al., 2020; Strokach & Kim, 2022). We divide the mainstream protein design methods into three categories: protein sequence design (Li et al., 2014; Wu et al., 2021; Pearce & Zhang, 2021; Ingraham et al., 2019; Jing et al., 2020; Tan et al., 2022; Gao et al., 2022a; Hsu et al., 2022; Dauparas et al., 2022; Gao et al., 2022b; O'Connell et al., 2018; Wang et al., 2018; Qi & Zhang, 2020; Strokach et al., 2020; Chen et al., 2019; Zhang et al., 2020; Anand & Achim, 2022), unconditional protein structure generation (Anand & Huang, 2018; Sabban & Markovsky, 2020; Eguchi et al., 2022; Wu et al., 2022a), and conditional protein design (Lee & Kim, 2022; Wang et al., 2022a; Trippe et al., 2022; Lai et al., 2022; Fu & Sun, 2022; Tischer et al., 2020; Anand & Achim, 2022; Luo et al.). Protein sequence design aims to discover protein sequences folding into the desired structure, and unconditional protein structure generation focus on generating new protein structures from noisy inputs. We are interested in conditional protein design and consider multiple constraints on the designed protein. For example, Wang's model (Wang et al., 2022a), SMCDiff (Trippe et al., 2022) and Tischer's model (Tischer et al., 2020) design the scaffold for the specified functional sites. ProteinSGM (Lee & Kim, 2022) mask short spans (< 8 residues) of different secondary structures in different structures and treats the

design task as a inpainting problem. CoordVAE (Lai et al., 2022) produces novel protein structures conditioned on the backbone template. RefineGNN (Jin et al., 2021), CEM (Fu & Sun, 2022), and DiffAb (Luo et al.) aim to generate the complementarity-determining regions of the antibody. We summarized protein design model in Table.6.

Table 6: Protein Design Models.

| Method | Input | Github |
|---|---|---|
| Unconditional protein structure generation | | |
| Anand's model (Anand & Huang, 2018) | Noise | PyTorch |
| RamaNet (Sabban & Markovsky, 2020) | Noise | TF |
| Ig-VAE (Eguchi et al., 2022) | Noise | PyTorch |
| FoldingDiff (Wu et al., 2022a) | Noise | PyTorch |
| Protein seqeunce design | | |
| GraphTrans Ingraham et al. (2019) | 3D Backbone | PyTorch |
| GVP (Jing et al., 2020) | 3D Backbone | PyTorch |
| GCA (Tan et al., 2022) | 3D Backbone | PyTorch |
| AlphaDesign (Gao et al., 2022a) | 3D Backbone | PyTorch |
| ESM-IF (Hsu et al., 2022) | 3D Backbone | PyTorch |
| ProteinMPNN (Dauparas et al., 2022) | 3D Backbone | PyTorch |
| PiFold (Gao et al., 2022b) | 3D Backbone | PyTorch |
| Conditional protein design | | |
| ProteinSGM (Lee & Kim, 2022) | Masked structures | – |
| Wang's model Wang et al. (2022a) | Functional sites | PyTorch |
| SMCDiff (Trippe et al., 2022) | Functional motifs | – |
| CoordVAE Lai et al. (2022) | Backbone Template | – |
| CEM Fu & Sun (2022) | CDR geometry | – |
| Tischer's model (Tischer et al., 2020) | Functional motifs | TF |
| Anand's model (Anand & Achim, 2022) | Multiple conditions | – |
| RefineGNN (Jin et al., 2021) | Antigen structure | PyTorch |
| DiffAb (Luo et al.) | Antigen structure | PyTorch |

## A.2 ALGORITHMS

The direction representation could be equivalently transformed to the angle-based one:

$$
\begin{cases}
\alpha_i^N = \arccos(-(\boldsymbol{d}_{i+1}^N)^T \boldsymbol{d}_i^C) \\
\alpha_i^A = \arccos(-(\boldsymbol{d}_i^A)^T \boldsymbol{d}_i^N) \\
\alpha_i^C = \arccos(-(\boldsymbol{d}_i^C)^T \boldsymbol{d}_i^A) \\
\beta_i^N = \mathrm{dihedral}(\boldsymbol{d}_i^A, \boldsymbol{d}_i^C, \boldsymbol{d}_{i+1}^N) \\
\beta_i^A = \mathrm{dihedral}(\boldsymbol{d}_i^C, \boldsymbol{d}_{i+1}^N, \boldsymbol{d}_{i+1}^A) \\
\beta_i^C = \mathrm{dihedral}(\boldsymbol{d}_i^N, \boldsymbol{d}_i^A, \boldsymbol{d}_i^C)
\end{cases}
\tag{17}
$$

where $\mathrm{dihedral}(\boldsymbol{v}_1, \boldsymbol{v}_2, \boldsymbol{v}_3)$ is defined in Alg.2 (Appendix).

---

**Algorithm 1** place$(\boldsymbol{x}_{i-1}, \alpha_i, \beta_i, \boldsymbol{d}_{i-1}, \boldsymbol{d}_{i-2}, r)$

---

1: **Input:** $\boldsymbol{x}_{i-1}, \alpha_i, \beta_i, \boldsymbol{d}_{i-1}, \boldsymbol{d}_{i-2}$
2: $\tilde{\boldsymbol{d}}_i = [-\cos\alpha_i, \cos\beta_i\sin\alpha_i, \sin\beta_i\sin\alpha_i]^T$
3: $R_i = [\boldsymbol{d}_{i-1}, (\boldsymbol{d}_{i-2} \times \boldsymbol{d}_{i-1}) \times \boldsymbol{d}_{i-1}, \boldsymbol{d}_{i-2} \times \boldsymbol{d}_{i-1}]$
4: $\boldsymbol{d}_i = R_i \tilde{\boldsymbol{d}}_i$
5: **Return:** $\boldsymbol{x}_{i-1} + r_i \boldsymbol{d}_i$

---

**Gradient Analysis** The geometric constants in Eq.2 equires to reduce the loss function of $\mathcal{L}(\|\boldsymbol{p}_i^A - \boldsymbol{p}_1^A\|)$. Lets rewrite $\{\boldsymbol{p}_1^A, \boldsymbol{p}_1^C, \boldsymbol{p}_2^N, \boldsymbol{p}_2^A, \boldsymbol{p}_2^C, \cdots, \boldsymbol{p}_i^N, \boldsymbol{p}_i^A\}$ as $\{\boldsymbol{x}_0, \boldsymbol{x}_1, \boldsymbol{x}_2, \boldsymbol{x}_3, \cdots, \boldsymbol{x}_{n-1}, \boldsymbol{x}_n\}$ and define $\boldsymbol{d}_i = (\boldsymbol{x}_i - \boldsymbol{x}_{i-1})/\|\boldsymbol{x}_i - \boldsymbol{x}_{i-1}\|$, $r_i = \|\boldsymbol{x}_i - \boldsymbol{x}_{i-1}\|$, $\boldsymbol{e} = (\boldsymbol{p}_i^A - \boldsymbol{p}_1^A)/\|(\boldsymbol{p}_i^A - \boldsymbol{p}_1^A)\|$. The original loss function could be reformulated as $\mathcal{L}(\sum_{i=1}^n r_i \boldsymbol{e}^T \boldsymbol{d}_i)$. When using Algorithm.1, the $\boldsymbol{d}_i$ is a function of $\boldsymbol{d}_{i-1}$ and $\boldsymbol{d}_{i-2}$. Denote $L = \sum_{i=1}^n r_i \boldsymbol{e}^T \boldsymbol{d}_i$, the gradient of $L$ w.r.t. $\boldsymbol{d}_i$ is:

$$
\frac{\partial\mathcal{L}}{\partial\boldsymbol{d}_i} = \frac{\partial\mathcal{L}}{\partial L}\left(\sum_{k=i}^n r_k \boldsymbol{e}^T \frac{\partial\boldsymbol{d}_k}{\partial\boldsymbol{d}_i}\right)
\tag{18}
$$

$$
\tag{19}
$$

We show the gradient computational graph in Fig.5.

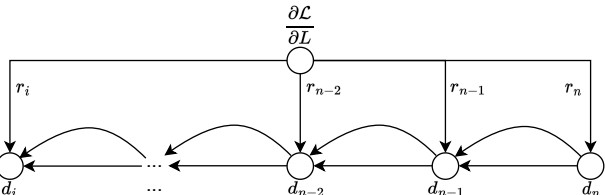

Figure 5: The gradient computational graph.

**Assumption A.1.** During the initial stage of training, the folding angles $\alpha_i$ and $\beta_i$ are iid random values, whose distributios are $\alpha_i \sim \mathcal{N}(\mu_\alpha, \sigma_\alpha^2)$ and $\beta_i \sim \mathcal{N}(\mu_\beta, \sigma_\beta^2)$, respectively.

**Lemma A.2.**

$$
\boldsymbol{u} \times \boldsymbol{v} = C(\boldsymbol{u})\boldsymbol{v} = \begin{bmatrix} 0 & -u_3 & u_2 \\ u_3 & 0 & -u_1 \\ -u_2 & u_1 & 0 \end{bmatrix} \begin{bmatrix} v_1 \\ v_2 \\ v_3 \end{bmatrix}
\tag{20}
$$

$$
C(\boldsymbol{u})\boldsymbol{v} = -[\boldsymbol{v}^T C(\boldsymbol{u})]^T
\tag{21}
$$

**Lemma A.3.**

$$
(\boldsymbol{a} \times \boldsymbol{b}) \times \boldsymbol{c} = (\boldsymbol{c} \cdot \boldsymbol{a})\boldsymbol{b} - (\boldsymbol{c} \cdot \boldsymbol{b})\boldsymbol{a}
$$
$$
\boldsymbol{a} \times (\boldsymbol{b} \times \boldsymbol{c}) = (\boldsymbol{a} \cdot \boldsymbol{c})\boldsymbol{b} - (\boldsymbol{a} \cdot \boldsymbol{b})\boldsymbol{c}
$$

**Compute $\frac{\partial \boldsymbol{d}_k}{\partial \boldsymbol{d}_i}$.** From Algorithm.1, we have:

$$
\begin{aligned}
\boldsymbol{d}_i &= R_i \tilde{\boldsymbol{d}}_i \\
&= -\cos\alpha_i \boldsymbol{d}_{i-1} + \cos\beta_i \sin\alpha_i (\boldsymbol{d}_{i-2} \times \boldsymbol{d}_{i-1}) \times \boldsymbol{d}_{i-1} + \sin\beta_i \sin\alpha_i \boldsymbol{d}_{i-2} \times \boldsymbol{d}_{i-1} \\
&= -\cos\alpha_i \boldsymbol{d}_{i-1} + \cos\beta_i \sin\alpha_i (\boldsymbol{d}_{i-2}^T \boldsymbol{d}_{i-1} \boldsymbol{d}_{i-1} - \boldsymbol{d}_{i-1}^T \boldsymbol{d}_{i-1} \boldsymbol{d}_{i-2}) + \sin\beta_i \sin\alpha_i \boldsymbol{d}_{i-2} \times \boldsymbol{d}_{i-1} \\
&= -\cos\alpha_i \boldsymbol{d}_{i-1} + \cos\beta_i \sin\alpha_i (-\cos\alpha_{i-1}\boldsymbol{d}_{i-1} - \boldsymbol{d}_{i-2}) + \sin\beta_i \sin\alpha_i \boldsymbol{d}_{i-2} \times \boldsymbol{d}_{i-1}
\end{aligned}
$$

Considering $\boldsymbol{d}_{i-2}^T \boldsymbol{d}_{i-1} = -\cos\alpha_{i-1}$ and $||\boldsymbol{d}_{i-1}|| = 1$, we have

$$
\begin{cases}
\frac{\partial \boldsymbol{d}_i}{\partial \boldsymbol{d}_{i-1}} = (-\cos\alpha_i - \cos\beta_i \sin\alpha_i \cos\alpha_{i-1})\boldsymbol{I} + \sin\beta_i \sin\alpha_i C(\boldsymbol{d}_{i-2}) \\
\frac{\partial \boldsymbol{d}_i}{\partial \boldsymbol{d}_{i-2}} = -\cos\beta_i \sin\alpha_i \boldsymbol{I} - \sin\beta_i \sin\alpha_i C(\boldsymbol{d}_{i-1})
\end{cases} \tag{22}
$$

The differential computation graph is shown in Fig.5, where we conclude that:

$$
\begin{cases}
\frac{\partial \boldsymbol{d}_n}{\partial \boldsymbol{d}_{n-1}} = (-\cos\alpha_n - \cos\beta_n \sin\alpha_n \cos\alpha_{n-1})\boldsymbol{I} + \sin\beta_n \sin\alpha_n C(\boldsymbol{d}_{n-2}) \\
\frac{\partial \boldsymbol{d}_n}{\partial \boldsymbol{d}_{n-2}} = -\cos\beta_n \sin\alpha_n \boldsymbol{I} - \sin\beta_n \sin\alpha_n C(\boldsymbol{d}_{n-1}) \\
\frac{\partial \boldsymbol{d}_n}{\partial \boldsymbol{d}_{n-3}} = \frac{\partial \boldsymbol{d}_n}{\partial \boldsymbol{d}_{n-2}} \frac{\partial \boldsymbol{d}_{n-2}}{\partial \boldsymbol{d}_{n-3}} + \frac{\partial \boldsymbol{d}_n}{\partial \boldsymbol{d}_{n-1}} \frac{\partial \boldsymbol{d}_{n-1}}{\partial \boldsymbol{d}_{n-3}} \\
\frac{\partial \boldsymbol{d}_n}{\partial \boldsymbol{d}_{n-4}} = \frac{\partial \boldsymbol{d}_n}{\partial \boldsymbol{d}_{n-3}} \frac{\partial \boldsymbol{d}_{n-3}}{\partial \boldsymbol{d}_{n-4}} + \frac{\partial \boldsymbol{d}_n}{\partial \boldsymbol{d}_{n-2}} \frac{\partial \boldsymbol{d}_{n-2}}{\partial \boldsymbol{d}_{n-4}} \\
\dots
\end{cases} \tag{23}
$$

**Case study** The Eq.23 is intractable. We consider the special case such as the beta-sheet, where the $\boldsymbol{d}_i$ follows the same primary direction $\boldsymbol{e}$. This allow us use $\boldsymbol{e}$ to approximate $\boldsymbol{d}_i$ and simplify the Eq.23 as:

$$
\begin{cases}
\frac{\partial \boldsymbol{d}_n}{\partial \boldsymbol{d}_{n-1}} \approx (-\cos\alpha_n - \cos\beta_n \sin\alpha_n \cos\alpha_{n-1})\boldsymbol{I} + \sin\beta_n \sin\alpha_n C(\boldsymbol{e}) \\
\frac{\partial \boldsymbol{d}_n}{\partial \boldsymbol{d}_{n-2}} \approx -\cos\beta_n \sin\alpha_n \boldsymbol{I} - \sin\beta_n \sin\alpha_n C(\boldsymbol{e}) \\
\dots
\end{cases} \tag{24}
$$

Denote $a_{\frac{n}{n-1}} = (-\cos\alpha_n - \cos\beta_n \sin\alpha_n \cos\alpha_{n-1})$ and $b_{\frac{n}{n-2}} = -\cos\beta_n \sin\alpha_n$, we have:

$$
\begin{cases}
\boldsymbol{e}^T \frac{\partial \boldsymbol{d}_n}{\partial \boldsymbol{d}_{n-1}} \approx a_{\frac{n}{n-1}} \boldsymbol{e}^T \\
\boldsymbol{e}^T \frac{\partial \boldsymbol{d}_n}{\partial \boldsymbol{d}_{n-2}} \approx b_{\frac{n}{n-2}} \boldsymbol{e}^T \\
\dots \\
\boldsymbol{e}^T \frac{\partial \boldsymbol{d}_n}{\partial \boldsymbol{d}_{n-k}} \approx a_{\frac{n-(k-1)}{n-k}} \boldsymbol{e}^T \frac{\partial \boldsymbol{d}_n}{\partial \boldsymbol{d}_{n-(k-1)}} + b_{\frac{n-(k-2)}{n-k}} \boldsymbol{e}^T \frac{\partial \boldsymbol{d}_n}{\partial \boldsymbol{d}_{n-(k-2)}} \\
\dots
\end{cases} \tag{25}
$$

**Projected Gradient** Lets compute the expected gradient value following the direction $\boldsymbol{e}$:

$$
\mathbb{E}[\boldsymbol{e}^T \frac{\partial \boldsymbol{d}_n}{\partial \boldsymbol{d}_{n-k}} \boldsymbol{e}] \approx \mathbb{E}[a_{\frac{n-(k-1)}{n-k}}]\mathbb{E}[\boldsymbol{e}^T \frac{\partial \boldsymbol{d}_n}{\partial \boldsymbol{d}_{n-(k-1)}} \boldsymbol{e}] + \mathbb{E}[b_{\frac{n-(k-2)}{n-k}}]\mathbb{E}[\boldsymbol{e} \frac{\partial \boldsymbol{d}_n}{\partial \boldsymbol{d}_{n-(k-2)}} \boldsymbol{e}] \tag{26}
$$

According to the assumption A.1, we know that $\mathbb{E}[a_{\frac{n-(k-1)}{n-k}}]$ and $\mathbb{E}[b_{\frac{n-(k-2)}{n-k}}]$ are constants. Therefore, we conclude

$$
\mathbb{E}[\boldsymbol{e}^T \frac{\partial \boldsymbol{d}_n}{\partial \boldsymbol{d}_{n-k}} \boldsymbol{e}] = K_1 t_1^k + K_2 t_2^k \tag{27}
$$

where $K_1, K_2, t_1, t_2$ are constants. When the value of $k$ becomes sufficiently large, the issue of vanishing or exploding gradients would arise.

### A.3 Additional Figures

**Algorithm 2** dihedral($\boldsymbol{v}_1, \boldsymbol{v}_2, \boldsymbol{v}_3$)

1: **Input:** $\boldsymbol{v}_1, \boldsymbol{v}_2, \boldsymbol{v}_3$
2: $\boldsymbol{n}_1 = \boldsymbol{v}_1 \times \boldsymbol{v}_2$
3: $\boldsymbol{n}_2 = \boldsymbol{v}_2 \times \boldsymbol{v}_3$
4: $x = (\boldsymbol{n}_1)^T \boldsymbol{n}_2$
5: $y = \boldsymbol{n}_1 \times \boldsymbol{n}_2$
6: **Return:** $\arctan \frac{y}{x}$

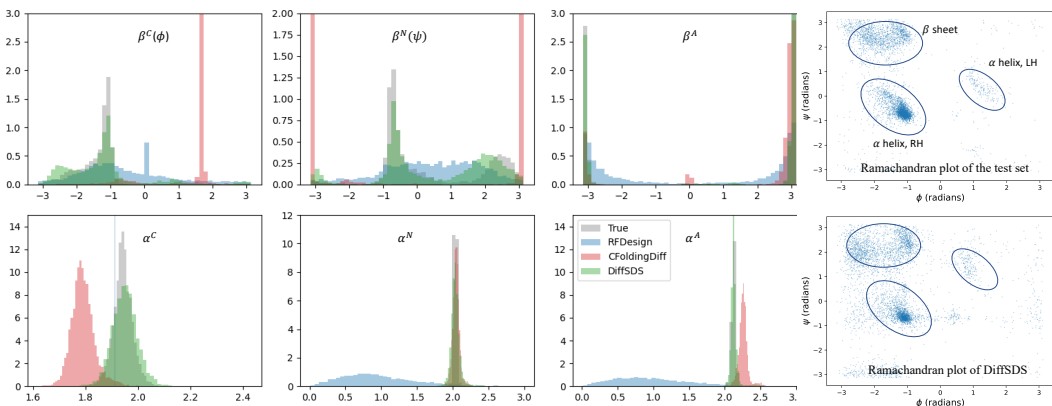

Figure 6: Angle distributions of various methods vs the test set distribution. DiffSDS produces the most similar angle distributions to those of the test set. The Ramachandran plots also show that DiffSDS can produce realistic structural distributions

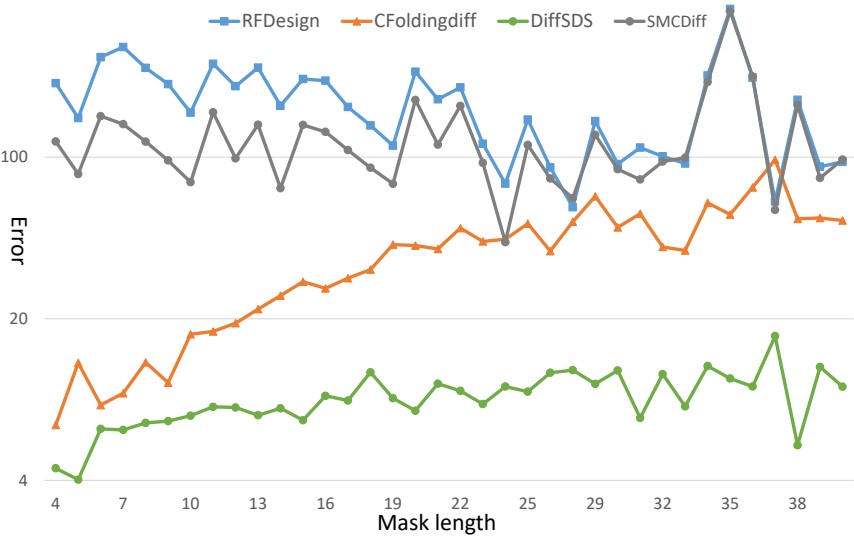

Figure 7: The connectivity error trend of different methods. SMCDiff and RFDesign perform poorly at all mask lengths, FoldingDiff's connectivity error increases with mask length, while DiffSDS performs steadily and consistently better than all baselines.

### A.4 DIFFUSION

**Forward process.**  We start from the standard diffusion process $x_0 \to x_1 \to \cdots \to x_T$, where the forward translation kernel from timestamp $s$ to $t$ is defined as $q(x_t|x_s) = \mathcal{N}(x_t; \alpha_{t|s}x_s, \sigma_{t|s}^2 I)$, $s \leq t$. Denote $\alpha_t = \alpha_{t|0}, \sigma_t = \sigma_{t|0}$, and $q(x_0|x_0) = \mathcal{N}(x_0; \alpha_0 x, \sigma_0^2 I), \alpha_0 = 1, \sigma_0 = 0$. We will show that $\alpha_{t|s} = \alpha_t/\alpha_s, \sigma_{t|s}^2 = \sigma_t^2 - \alpha_{t|s}^2 \sigma_s^2$.

*Proof.*

$$
\begin{aligned}
q(x_t|x_s) =& \mathcal{N}(x_t; \alpha_{t|s}x_s, \sigma_{t|s}^2 I) \\
\Rightarrow x_t =& \alpha_{t|t-1}x_{t-1} + \sigma_{t|t-1}\epsilon_{t-1} & x_t \sim q(x_t|x_{t-1}) \\
=& \alpha_{t|t-1}(\alpha_{t-1|s}x_s + \sigma_{t-1|s}\epsilon_s) + \sigma_{t|t-1}\epsilon_{t-1} & x_{t-1} \sim q(x_{t-1}|x_s) \\
=& \alpha_{t|t-1}\alpha_{t-1|s}x_s + \alpha_{t|t-1}\sigma_{t-1|s}\epsilon_s + \sigma_{t|t-1}\epsilon_{t-1} \\
=& (\alpha_{t|t-1}\alpha_{t-1|s})x_s + \sqrt{\alpha_{t|t-1}^2 \sigma_{t-1|s}^2 + \sigma_{t|t-1}^2}\hat{\epsilon}_s \\
=& \alpha_{t|s}x_s + \sigma_{t|s}\epsilon_s & x_t \sim q(x_t|x_s)
\end{aligned}
\tag{28}
$$

We conclude that $\alpha_{t|s} = \alpha_{t|t-1}\alpha_{t-1|s}$ and $\sigma_{t|s} = \sqrt{\alpha_{t|t-1}^2 \sigma_{t-1|s}^2 + \sigma_{t|t-1}^2}$.

For $\alpha_{t|s} = \alpha_{t|t-1}\alpha_{t-1|s}$,

$$
\begin{aligned}
\alpha_{t|s} &= \alpha_{t|t-1}\alpha_{t-1|s} \\
&= \alpha_{t|t-1}\alpha_{t-1|t-2}\alpha_{t-2|s} \\
&\cdots \\
&= \prod_{i=s+1}^{t} \alpha_{i|i-1} \\
&= \prod_{i=s+1}^{t} \frac{\alpha_i}{\alpha_{i-1}}
\end{aligned}
\tag{29}
$$

For $\sigma_{t|s} = \sqrt{\alpha_{t|t-1}^2 \sigma_{t-1|s}^2 + \sigma_{t|t-1}^2}$, let $s = 0$, then $\sigma_{t|t-1}^2 = \sigma_t^2 - \alpha_{t|t-1}^2 \sigma_{t-1}^2$. Therefore,

$$
\begin{aligned}
\sigma_{t|s}^2 &= \alpha_{t|t-1}^2 \sigma_{t-1|s}^2 + \sigma_{t|t-1}^2 \\
&= \alpha_{t|t-1}^2 (\alpha_{t-1|t-2}^2 \sigma_{t-2|s}^2 + \sigma_{t-1|t-2}^2) + \sigma_{t|t-1}^2 \\
&= \alpha_{t|t-1}^2 \alpha_{t-1|t-2}^2 \sigma_{t-2|s}^2 + \alpha_{t|t-1}^2 \sigma_{t-1|t-2}^2 + \sigma_{t|t-1}^2 \\
&= \alpha_{t|t-2}^2 \sigma_{t-2|s}^2 + \alpha_{t|t-1}^2 \sigma_{t-1|t-2}^2 + \alpha_{t|t}^2 \sigma_{t|t-1}^2 \\
&= \sum_{k=s+1}^{t} \alpha_{t|k}^2 \sigma_{k|k-1}^2 \\
&= \sum_{k=s+1}^{t} \alpha_{t|k}^2 (\sigma_k^2 - \alpha_{k|k-1}^2 \sigma_{k-1}^2) & \text{Apply } \sigma_{t|t-1}^2 = \sigma_t^2 - \alpha_{t|t-1}^2 \sigma_{t-1}^2 \\
&= \sum_{k=s+1}^{t} \alpha_{t|k}^2 \sigma_k^2 - \sum_{k=s+1}^{t} \alpha_{t|k-1}^2 \sigma_{k-1}^2 \\
&= \alpha_{t|t}^2 \sigma_t^2 - \alpha_{t|s}^2 \sigma_s^2 \\
&= \sigma_t^2 - \alpha_{t|s}^2 \sigma_s^2
\end{aligned}
\tag{30}
$$

□

**Reverse process.** As to the backward process $x_T \to x_{T-1} \to \cdots \to x_0$, the neural network aims to maximize $q(x_s|x_t, x_0) = \mathcal{N}(z_s; \hat{\mu}_s, \hat{\sigma}_s^2 I)$, where

$$\begin{cases} \hat{\sigma}_s = \frac{\sigma_{t|s}\sigma_s}{\sigma_t} \\ \hat{\mu}_s = \frac{1}{\alpha_{t|s}}x_t - \frac{\alpha_{t|s}}{\sigma_t}\epsilon_t \end{cases} \tag{31}$$

*Proof.* Recall that $\mathcal{N}(z; \mu, \sigma I) \propto \exp\left(-\frac{||z-\mu||^2}{2\sigma^2}\right)$, the backward translation could be derived by Bayes' Theorem:

$$\begin{aligned} q(x_s|x_t, x_0) &= \frac{q(x_t|x_s)q(x_s|x_0)}{q(x_t|x_0)} \\ &\propto \exp\left[-\frac{1}{2}\left(\frac{||x_t - \alpha_{t|s}x_s||^2}{\sigma_{t|s}^2} + \frac{||x_s - \alpha_s x_0||^2}{\sigma_s^2} - \frac{||x_t - \alpha_t x_0||^2}{\sigma_t^2}\right)\right] \end{aligned} \tag{32}$$

from which we can derive $q(z_s|z_t, x_0) = \mathcal{N}(z_s; \hat{\mu}_s, \hat{\sigma}_s^2 I)$, and

$$\begin{cases} \frac{1}{\hat{\sigma}_s^2} = \frac{\alpha_{t|s}^2}{\sigma_{t|s}^2} + \frac{1}{\sigma_s^2} \\ -2\frac{\hat{\mu}_s}{\hat{\sigma}_s^2} = -2\frac{x_t\alpha_{t|s}}{\sigma_{t|s}^2} - 2\frac{\alpha_s x_0}{\sigma_s^2} \end{cases} \tag{33}$$

finally

$$\begin{cases} \hat{\sigma}_s = \frac{\sigma_{t|s}\sigma_s}{\sigma_t} \\ \hat{\mu}_s = \frac{1}{\alpha_{t|s}}x_t - \frac{\alpha_{s|t}\sigma_{t|s}^2}{\sigma_t}\epsilon_t \end{cases} \tag{34}$$

Note that

$$\begin{aligned} -2\frac{\hat{\mu}_s}{\hat{\sigma}_s^2} &= -2\frac{(x_t)\alpha_{t|s}}{\sigma_{t|s}^2} - 2\frac{(\alpha_s x_0)}{\sigma_s^2} \\ \hat{\mu}_s &= \frac{(x_t)\alpha_{t|s}}{\sigma_{t|s}^2}\left(\frac{\sigma_{t|s}\sigma_s}{\sigma_t}\right)^2 + \frac{(\alpha_s x_0)}{\sigma_s^2}\left(\frac{\sigma_{t|s}\sigma_s}{\sigma_t}\right)^2 \\ \hat{\mu}_s &= \frac{(x_t)\alpha_{t|s}\sigma_s^2}{\sigma_t^2} + \frac{(\alpha_s x_0)\sigma_{t|s}^2}{\sigma_t^2} \\ \hat{\mu}_s &= \frac{(x_t)\alpha_{t|s}\sigma_s^2}{\sigma_t^2} + \alpha_s\frac{x_t - \sigma_t\epsilon_t}{\alpha_t}\frac{\sigma_{t|s}^2}{\sigma_t^2} \\ \hat{\mu}_s &= \left(\frac{\alpha_{t|s}\sigma_s^2}{\sigma_t^2} + \frac{\alpha_s\sigma_{t|s}^2}{\alpha_t\sigma_t^2}\right)x_t - \frac{\alpha_s\sigma_{t|s}^2}{\alpha_t\sigma_t}\epsilon_t \\ \hat{\mu}_s &= \frac{1}{\alpha_{t|s}}x_t - \frac{\alpha_{s|t}\sigma_{t|s}^2}{\sigma_t}\epsilon_t \end{aligned} \tag{35}$$

$\square$

A.5 ABLATION

**Objective&Setting.** We conduct ablation experiments to investigate the impact of conditions and constraints on protein backbone inpainting. Specifically, we show how these factors affect the validation losses, including angle loss, length loss, and overlapping loss.

**Results & Analysis.** As shown in Fig.8, Fig.9 and Tab.7, we find that: (1) the length condition and the sequence condition contribute to the reduction of $\mathcal{L}_{len}$ and $\mathcal{L}_{sim}$, respectively. This phenomenon suggests that the model can learn to generate structures with a predetermined length and that residue types can facilitate learning backbone angles. (2) Explicitly imposing geometric constraints on the model is necessary. If the constraints are removed, the $\mathcal{L}_{len}$ loss is difficult to reduce, and the $\mathcal{L}_{overlap}$ even increases, indicating the model could not generate geometrically reasonable structures. Fortunately, all these drawbacks could be eliminated by imposing geometric loss on the introduced direction space.

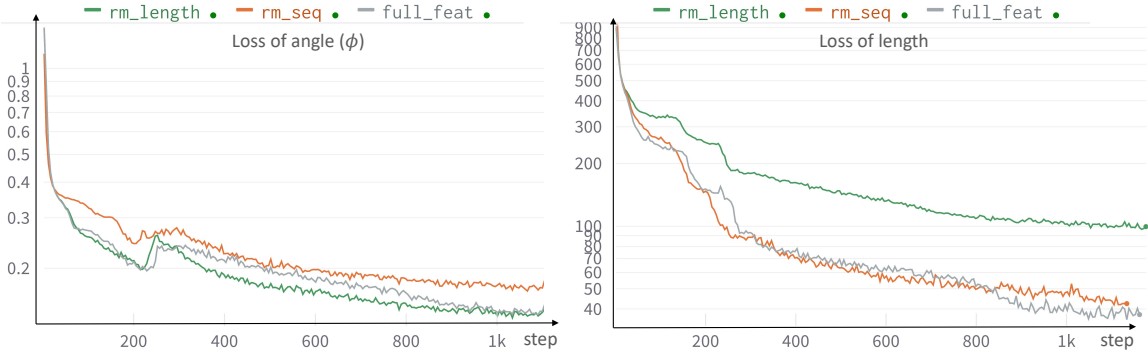

Figure 8: Ablation of conditions. We remove length condition or residue type embedding from the baseline (full_feat) model, resulting in rm_length and rm_seq, respectively. We show the loss curves for angle ($\phi$) and length on the validation set, revealing the effect of the conditions.

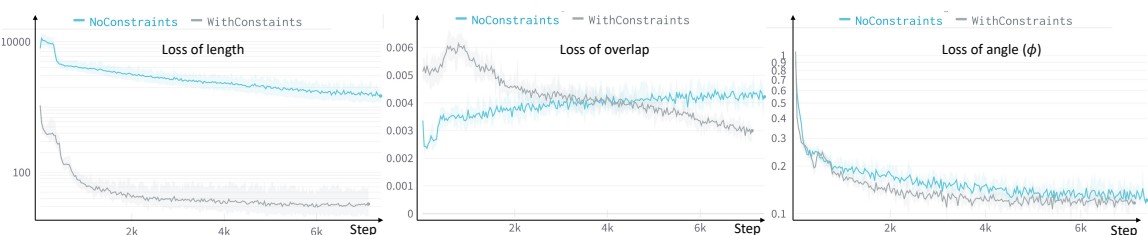

Figure 9: Ablation of constraints. We show the validation loss curves in the constrained and unconstrained cases. The unconstrained case means that $\mathcal{L}_{len}$ and $\mathcal{L}_{overlap}$ are not imposed on the model during training.

| Name | scTM |
|---|---|
| DiffSDS | 0.55 |
| w/o constraints | 0.53 |
| w/o sequence | 0.47 |

Table 7: Ablations of adding constraints and sequence features.

## A.6 MORE DISCUSSIONS

**Why does CFoldingDiff works poorly?** We analyzed the training and validation curves of CFoldingDiff and DiffSDS for the angles $\phi$ and $\psi$. Our results show that CFoldingDiff is prone to overfitting without applying geometric constraints. Although the training loss of CFoldingDiff decreased better than that of DiffSDS, the validation loss remained high. We hypothesize that the geometric constraint loss may act as an auxiliary regularizer, facilitating the model's convergence towards a good global minimum.

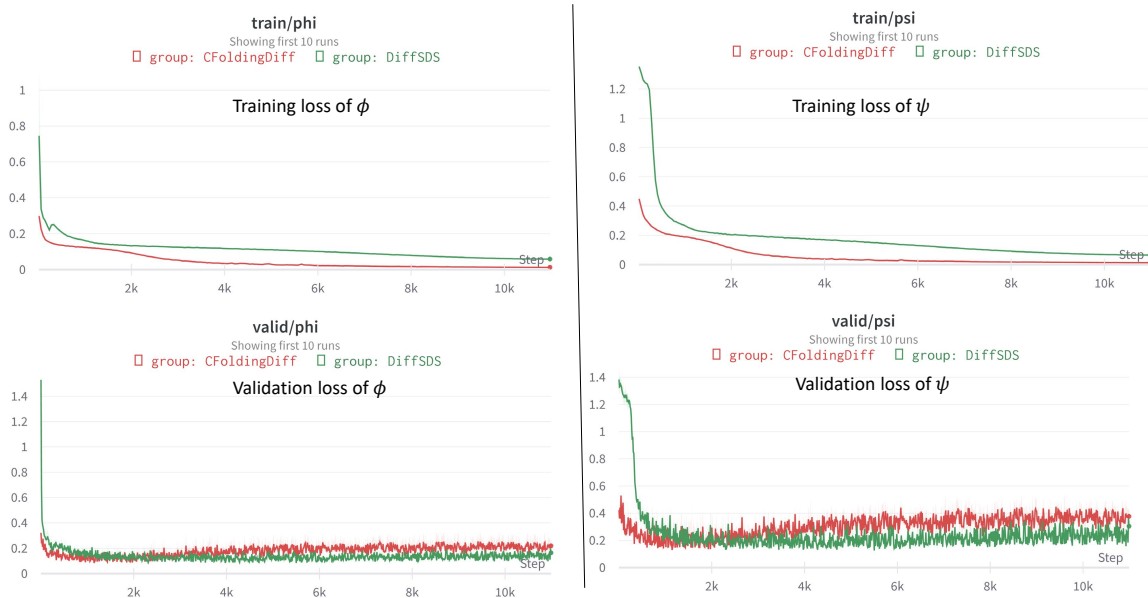

Figure 10: Training and validation curves of CFoldingDiff and DiffSDS.

**Why do RFDesign and SMCDiff work poorly?** As shown in the following tables, we argue that structural breaks due to the lack of structural constraints would corrupt the input features of ESM-IF for SMCDiff and RFDesign, producing poor sequence design and ultimately poor scTM scores.

| | Connectivity error ($\downarrow$) | | |
|---|---|---|---|
| mask length | $<10$ | 10-15 | $>15$ |
| SMCDiff | 113.62 | 107.32 | 154.52 |
| RFDesign | 218.36 | 146.24 | 159.17 |
| CFoldingDiff | 14.77 | 42.65 | 59.08 |
| DiffSDS | **6.93** | **9.93** | **10.61** |

Table 8: connectivity error of different methods.

| | scTM>0.5 ($\uparrow$) | | | scTM ($\uparrow$) | |
|---|---|---|---|---|---|
| | All | len$\leq$70 | len>70 | Mean | Median |
| SMCDiff | 30/378 | 12/148 | 18/230 | 0.36 | 0.34 |
| RFDesign | 178/378 | 65/148 | 113/230 | 0.51 | 0.48 |
| CFoldingDiff | 217/378 | 81/148 | 130/230 | 0.54 | 0.53 |
| DiffSDS | **231/378** | **88/148** | **143/230** | **0.56** | **0.55** |
| Relative Gain | 6.5% | 8.6% | 10% | 3.7% | 3.8% |

Table 9: Designability of different methods.

**Other Comments**    Since SMCDiff only generates $C_\alpha$, we are unable to calculate the folding angles and pairwise residue distances considering $C$, $C_\alpha$, and $O$, and thus cannot provide Rosetta energy and spatial interaction metrics.