# OpenReview forum: "DiffSDS: A geometric sequence diffusion model for protein backbone inpainting"
_ICLR.cc/2024/Conference — ICLR 2024 Conference Withdrawn Submission_

### Official Review · Reviewer_qjdQ · 2023-10-30

**Soundness:** 2 fair
**Presentation:** 2 fair
**Contribution:** 1 poor
**Rating:** 3
**Confidence:** 5

**Summary:**

The paper tried to solve the inpainting problem for protein structures by starting from any input of initial and final subchains consisting of given (unmasked) atoms (three main atoms CA, C, N from each residue) with fixed positions.

**Strengths:**

The authors should be highly praised for studying important real objects such as proteins.

The paper is generally well-written and contains enough details that helped understand the difficulties.

The "problem definition" in section 1.1 is a good start towards a well-posed problem but misses important assumptions and quality measures.

**Weaknesses:**

For the majority of potential inputs in "problem definition", the output should be: impossible to find the missing atoms. Even if we start from a real protein and remove only one amino acid residue, the remaining residues can not be in arbitrary positions. We can even rigidly move the resulting subchains by translation or rotation away from each other, which makes a potential reconstruction impossible in almost all cases.

A well-posed problem should include a proper metric for quality of the reconstruction. Unfortunately, almost all widely used similarity distances (including LDDT and 1-TM score) between proteins fail the metric axioms (en.wikipedia.org/wiki/Metric_space).

If the first axiom fails, distance 0 doesn't guarantee that the given structures are equivalent. If the triangle inequality fails, Rass et al (arxiv:2211.03674) proved that the popular clustering algorithms such as k-means and DBSCAN can output any predetermined clusters.

Even with all these additions to the problem statement, there is no guarantee of a unique solution because proteins are dynamic objects in living organisms. Hence their structures in the Protein Data Bank (PDB) or other datasets are only still images of continuous movies.

The weakness of the direction-based representation, which is claimed to be novel in section 3.1, is the reliance on "fixed" bond lengths, which can actually vary in wide ranges, and the non-invariance of direction vectors under rotations.

**Questions:**

How justified are the thresholds (0.5 for scTM, 70 for len, and many others for mask length)? What happens with the results if these thresholds are changed, even if slightly changed?

How many CPU hours and hidden parameters were used in the experiments?

---

### Official Review · Reviewer_YgsM · 2023-11-01

**Soundness:** 3 good
**Presentation:** 3 good
**Contribution:** 2 fair
**Rating:** 5
**Confidence:** 4

**Summary:**

In the task of protein backbone inpainting, the authors introduce a novel model named DiffSDS. This model's architecture encompasses both an encoder and a decoder, uniquely enriched with a hidden atomic direction space (ADS) layer embedded within the encoder. This ADS layer plays a pivotal role in translating backbone angles into equivariant direction vectors. The decoder then takes on the responsibility of reclaiming these vectors and reverting them to their original backbone angles. The authors highlight that prior angle-based techniques, aimed at depicting the spatial relationships of protein backbone atoms, might grapple with issues like gradient explosion or vanishing. To tackle this, in their experiments, they recalibrated a CFoldingDiff model specifically tailored for the protein inpainting task.

**Strengths:**

1. The authors propose a unique encoder-decoder structure tailored exclusively for the protein backbone inpainting task, setting a fresh benchmark in the process.

2. They pioneer a directional representation, adept at computing geometric features efficiently, all while preserving a minimalistic degree of freedom.
3. The strategic inclusion of an extra loss component aids in stabilizing the decrement of training loss, subsequently bolstering the overall efficacy of the model.
4. Through their experimentation, the authors reveal that utilizing the angle-centric approach for direct training might encounter issues like gradient explosion or vanishing.

**Weaknesses:**

1. The author suggests the integration of a hidden atomic direction space (ADS) layer to bolster the model's efficacy. Yet, the specifics of the ADS layer, particularly its architecture, remain undisclosed, leading to ambiguities surrounding the angle-to-direction vector conversion process.

2. The authors introduce an inventive direction representation. However, its application is restricted solely to their designated latent direction space. As such, for protein backbone inpainting, there remains a necessity to calculate the angles for reconstruction.

3. In the presented experiments, while their model outperforms another angle-centric model, namely CFoldingDiff, it doesn't quite match up to frame-centric models like RFDiffusion for most of the outlined tasks.

4. Figure 6 has been relegated to the supplementary materials and isn't seamlessly integrated within the primary text, causing a fragmented flow of annotations.

5. Frame-based models are not specifically designed for the protein backbone inpainting task, which may render the comparison with such models potentially unfair for this application.

**Questions:**

1. The sidechain of a protein plays a significant role in determining its conformation. In the authors' experiments, it is not clear whether the comparison criteria consider the position of the protein's sidechain. If the sidechain position is considered, one might question how the authors determine the position of the sidechain, as this information is crucial for a comprehensive understanding of their approach.

2. What the architecture of the ADS layer, and how convert the angles into the direction vectors.

3. With respect to the Connectivity and non-Overlapping experiments, is there a possibility that a pronounced connectivity error might negate the significance of the non-overlapping score? Given that the test score for RFDiffusion appears to align closely with the authentic value, could this suggest an elevated connectivity level?

---

### Official Review · Reviewer_9CLX · 2023-11-01

**Soundness:** 3 good
**Presentation:** 3 good
**Contribution:** 2 fair
**Rating:** 3
**Confidence:** 3

**Summary:**

In the field of protein backbone inpainting with diffusion model, the authors proved that applying geometric constraints to the angle space would result in gradient vanishing or exploding, and then introduced a new method as remedy.
By adding a linear projection layer to transform angle-based representation into direction representation, and a decoder to transform direction representation back to angle-based representation, the authors utilized a dual-space diffusion model. Therefore, geometric constraints can be imposed on direction-based representation in the model. Meanwhile, the model designed a Transformer-based architecture for invariant data, such as angle-based representation for protein.

**Strengths:**

The paper offered theoretical proof of the occurrence of gradient vanishing or exploding when geometric constraints are applied to angle space representation, which gave the community an insight into the mechanism that hinders the wider use of angle representation. This could help the improvement of the protein structure angle representation approach in the future.

**Weaknesses:**

1. The motivation of the inpainting task itself is not clear to me. Why protein backbone inpainting is an important task? Is it practically meaningful and important?
2. The experimental results do not support the authors' claim that the proposed method "outperforms" or "provides competitive results" (compared to frame-based methods)  to existing methods. For instance, a clear performance reduction can be observed when comparing RFDiffusion and DIffSDS.
3. The connectivity evaluation is not persuasive to me. Many protein local structures are not rigid, especially at the loop/turn region and the endpoint of a protein sequence. The authors might provide additional justification on why it is problematic for a prediction to generate a different structure with the crystal structure at these positions.
4. The ADS layer in the Transformer requires additional investigation. More investigations could be conducted to validate the sufficient expressivity of a single linear transformation layer in transforming angle representation to direction representation.
5. An ablation study is expected to investigate how the accuracy of the ADS layer will influence the performance of a diffusion model so that one can verify the current choice of ADS layer is optimal.
6. Instead of simply calculating the overlapping loss only between masked and unmasked structures, overlapping in masked structures should be considered as well.
7. None of the experimental results are reported with a confidence interval, making the outperformance lack statistical significance.

**Questions:**

1. Eq(1): "endpoints $p_{s-1}^N$ and $p_{e+1}^N$: typo? should be $p_{s-1}^N$ and $p_{e+1}^N$.
2. "non-overlapping": Why the designed structure should not overlap with the native structure? Is it for novelty consideration?
3. There are so many relevant research or models for protein sequence generation. Why only a few of them were compared in Experiments?
4. Why is it that in Section 2 AlphaFold2 was cited as (the only) frame-based method for protein structure generation, while in Section (e.g., Table 1) RFDiffusion was selected as a baseline method?